# Relationships between Circulating Sclerostin, Bone Marrow Adiposity, Other Adipose Deposits and Lean Mass in Post-Menopausal Women

**DOI:** 10.3390/ijms24065922

**Published:** 2023-03-21

**Authors:** Marion Courtalin, Nicolas Bertheaume, Sammy Badr, Alexandrine During, Daniela Lombardo, Valérie Deken, Bernard Cortet, Aline Clabaut, Julien Paccou

**Affiliations:** 1Department of Rheumatology, University of Lille, 59000 Lille, France; 2Laboratory MABlab ULR 4490, 59000 Lille, France; 3Department of Radiology, University of Lille, 59000 Lille, France; 4METRICS—Evaluation des Technologies de Santé et des Pratiques Médicales, University of Lille, 59000 Lille, France

**Keywords:** bone marrow adiposity, sclerostin, Dixon method, MRI, post-menopausal women, osteoporosis, lean mass, adipose deposit

## Abstract

Sclerostin is a Wnt signaling pathway inhibitor that negatively regulates bone formation. Bone-marrow-derived stromal cell (BMSC) differentiation is influenced by the Wnt pathway, leading to the hypothesis that higher levels of sclerostin might be associated with an increase in bone marrow adiposity (BMA). The main purpose of this study was to determine whether a relationship exists between circulating sclerostin and BMA in post-menopausal women with and without fragility fractures. The relationships between circulating sclerostin and body composition parameters were then examined. The outcomes measures included vertebral and hip proton density fat fraction (PDFF) using the water fat imaging (WFI) MRI method; DXA scans; and laboratory measurements, including serum sclerostin. In 199 participants, no significant correlations were found between serum sclerostin and PDFF. In both groups, serum sclerostin was correlated positively with bone mineral density (R = 0.27 to 0.56) and negatively with renal function (R = −0.22 to −0.29). Serum sclerostin correlated negatively with visceral adiposity in both groups (R = −0.24 to −0.32). Serum sclerostin correlated negatively with total body fat (R = −0.47) and appendicular lean mass (R = −0.26) in the fracture group, but not in the controls. No evidence of a relationship between serum sclerostin and BMA was found. However, serum sclerostin was negatively correlated with body composition components, such as visceral adiposity, total body fat and appendicular lean mass.

## 1. Introduction

Sclerostin is a glycoprotein that is mainly secreted by osteocytes. It is also a soluble inhibitor of the canonical Wnt signaling pathway that negatively regulates bone formation by repressing osteoblast differentiation and proliferation [1,2,3]. In healthy adults, circulating sclerostin levels are correlated with age, sex, bone mineral density (BMD) and renal function [4,5].

Bone-marrow-derived stromal cell (BMSC) differentiation is influenced by the Wnt pathway, leading to the hypothesis that higher levels of sclerostin might be associated with an increase in bone marrow adiposity (BMA) [6]. The only previous clinical investigation that is available on the relationship between circulating sclerostin and BMA was conducted on the Iceland Age, Gene/Environment Susceptibility (AGES)-Reykjavik cohort [7]. In that study, conducted on older participants, circulating sclerostin was found to be positively associated with BMA in men, but not in women [7].

In vitro data on human BMSCs (hBMSCs) are insufficient to clearly determine the effect of sclerostin on bone marrow adipocytes (BMAds). The data from animal studies seem to support a proadipogenic effect of sclerostin on BMSCs at the expense of osteoblastogenesis [8]. Several animal models (sclerostin knock-out or treatment with an anti-sclerostin antibody) demonstrated that a decrease in BMA is associated with an improvement in bone parameters [8].

The separate effects of fat mass and lean mass on bone remain controversial, and specific fat deposits appear to have different relationships with bone. Moreover, investigations on the association between these components and circulating sclerostin are limited in number. In a preclinical analysis, high-fat diet (HFD) mice treated with an anti-sclerostin antibody exhibited a significant reduction in fat mass, suggesting that sclerostin may interfere with the communication between bone and adipose tissue [9]. Furthermore, in that study, the anti-sclerostin antibody was found to have no effect on lean mass [9]. Regarding body composition in humans, conflicting results were reported. A positive association between circulating sclerostin levels and total body fat (TBF) was reported in men, but not in women [7]. On the other hand, no association was found between circulating sclerostin levels and lean mass, either in men or women [7]. In contrast, the authors of another cross-sectional study found a negative association between serum sclerostin levels and lean mass in men [10].

Currently, knowledge of the interactions between bone, fat and muscle, and the role of circulating sclerostin in those interactions, is limited. Importantly, given that (i) anti-sclerostin antibodies are in the vanguard of treatment for post-menopausal osteoporosis, (ii) the role of bone marrow adipose tissue (BMAT) in the pathophysiology of the disease is increasingly acknowledged, and (iii) it is plausible to suggest that anti-sclerostin antibodies have an effect on body composition changes, data on the effects of anti-sclerostin antibodies on fat and muscle are expected to have important clinical implications [11].

The main purpose of this study was to determine whether a relationship exists between circulating sclerostin and BMA in post-menopausal women with and without fragility fractures. The relationships between circulating sclerostin and body composition parameters were then examined.

## 2. Results

### 2.1. Baseline Characteristics

An overview of the ADIMOS participants’ baseline characteristics and the differences between cases and controls is shown in Table 1. The mean age of the 199 participants was 59.0 ± 6.3 years. The controls (*n* = 99) were significantly younger than the cases (*n* = 100) (*p* < 0.001). When anthropometric measurements were considered, the controls were found to be significantly taller (*p* = 0.021) and heavier (*p* = 0.03). The risk factors for osteoporosis in both groups were comparable. The CCI was lower in the control group (*p* < 0.001). The controls had significantly lower 25(OH) D (*p* = 0.03) compared with the cases. In the cases, 52 women with at least one clinical vertebral fracture (median [min–max]: 1 [1–4]) and 48 women with non-vertebral fractures (18 distal forearm/wrist fractures, 14 hip fractures, 10 pelvis fractures, 5 proximal humerus fractures and 1 rib fracture) were included.

In this ancillary investigation, 198 samples were analyzed, as serum was not collected for one case. No significant differences in circulating sclerostin, leptin and adiponectin were found between cases and controls.

The controls had significantly higher BMDs than the cases at all sites (*p* < 0.001 for all). Further to the BMD testing, 43 cases were found to have osteoporosis (T-score ≤ −2.5) compared with 11 controls. No significant differences in body composition parameters were found between the cases and controls.

No significant differences in the PDFF were found between participants with and without fractures. Although the lumbar spine PDFF was higher in the cases than in the controls, the difference was not statistically significant (mean (SD): 59.1% (9.6) versus 56.6% (9.4), *p* = 0.06).

### 2.2. Correlations between Serum Sclerostin and Parameters of Interest

The correlations between serum sclerostin and the parameters of interest are shown in Table 2. In both groups, the femoral neck, total hip and lumbar spine BMDs correlated positively with circulating sclerostin (R = 0.27–0.56, *p* < 0.001 for all), suggesting higher serum sclerostin levels in post-menopausal women with higher BMDs, regardless of the fracture status. No significant correlations were found between circulating sclerostin, age, leptin and adiponectin. A significant negative correlation was found between serum sclerostin and eGFR in both groups (cases: R = −0.22, *p* = 0.03; controls: R = −0.29, *p* = 0.003), suggesting that serum sclerostin levels were higher in participants with a lower eGFR. A significant negative correlation was found between circulating sclerostin and serum PTH in the cases (R = −0.23 *p* = 0.02), but not in the controls.

### 2.3. Correlations between Circulating Sclerostin and PDFF (%)

In both the cases and controls, no significant correlations were found between circulating sclerostin and lumbar spine, femoral neck and femoral diaphysis PDFF, either before or after adjusting for age, eGFR and BMD (Table 3).

### 2.4. Correlations between Circulating Sclerostin and Body Composition Parameters

After adjusting for age, eGFR and lumbar spine BMD, serum sclerostin was negatively associated with VAT in both groups (R = −0.32, *p* = 0.002 in the cases; R = −0.24, *p* = 0.02 in the controls), and negatively correlated with TBF in the cases (R = −0.47, *p* < 0.001), but not in the controls (Table 4). Finally, serum sclerostin correlated negatively with ALM in the cases (R = −0.26, *p* = 0.01), but not in the controls.

## 3. Discussion

The main and novel finding of this analysis conducted on post-menopausal women was that no association was found between circulating sclerostin levels and BMA. Moreover, serum sclerostin levels correlated negatively with VAT in both groups, and with TBF and ALM in the cases, but not in the controls.

Examining and understanding the connection between BMAT and bone health recently emerged as an exciting area of research. BMAT is unique because it is the only tissue where BMAds and bone cells are juxtaposed. As first noted by Meunier [12], increased BMAT is seen in osteoporosis. Since then, increased interest in BMAT has led to several important findings [13,14]. However, data on the relationship between BMAT and circulating sclerostin are still scarce.

Positive correlations were found between circulating sclerostin levels and BMD measurements, which is in line with previous studies [5,15,16,17]. One proposed explanation for this positive association between sclerostin and BMD suggests that circulating sclerostin levels depend on the number of osteocytes, which is theoretically proportional to the total amount of bone, and on their activity. Higher BMD could result in more osteocytes and, therefore, higher circulating sclerostin levels [17].

Serum sclerostin levels did not correlate with age. This finding contrasts with those reported in previous observational studies, in which significant positive associations were found [5,16]. This discrepancy may be related to differences in population characteristics and, notably, the relatively narrow age range of our cohort.

A negative correlation between circulating sclerostin levels and eGFR was found. This correlation was already reported in several studies [4,18]. Circulating sclerostin levels are generally higher in CKD patients [19,20]. However, whether this is due to higher levels of osteocyte production, lower eGFR levels or both is not yet clear [21,22]. PTH was negatively associated with circulating sclerostin levels. This finding was consistently reported in previous studies [16,23,24] and suggests that PTH is probably one of the regulators of sclerostin secretion in post-menopausal women [25].

The close relationship between bone and fat suggests that there might be an association between sclerostin and adipokines (i.e., leptin and adiponectin). Little data are currently available and only one investigation, conducted by E. Grethen et al., suggested that leptin concentration could be a predictor of sclerostin concentrations in people with obesity [26]. No previous investigations were conducted on the relationship between adiponectin and sclerostin in adults. In this analysis, no relationship between circulating sclerostin levels and adipokines was found.

A few studies suggested that sclerostin may be associated with BMA [7,8,27]. In early 2010, Ma et al. reported that lumbar spine BMA measured using MRI and magnetic resonance spectroscopy (MRS) was greater in elderly men (*n* = 115) with higher serum sclerostin levels in models adjusted for age, BMI and diabetes, but not in women (*n* = 134) [7]. There were a few limitations to that study, such as the age of the population, which included only older white adults (mean age 79 years), and the fact that the analysis was limited to BMA measurement at the lumbar spine. Additionally, the factors used for adjustment were different from those used in our study and did not include BMD, which may have influenced the results. Furthermore, there are no previous studies on BMA and circulating sclerostin levels in adults that used PDFF measurements at the proximal hip.

Moreover, in another study conducted by H. Fairfield et al., using a sclerostin knock-out mouse model, the authors reported a significant decrease in proximal and distal tibia BMA (quantified by μCT) in sclerostin knock-out mice compared with wild-type mice [8]. When sclerostin-neutralizing antibody (Scl-Ab) was administered to wild-type mice, a decrease in the tibia and femur BMA (significantly fewer and smaller BMAds) was also observed [8]. Similarly, S. Costa et al. also reported that Scl-Ab decreases BMA in rats, primarily by increasing trabecular bone, thereby restricting the capacity for BMAd accumulation [28]. Interestingly, in a study that was already discussed, H. Fairfield et al. reported that hBMSCs treated with sclerostin in vitro appear to have greater expression of adipogenic genes (PPARɣ and CEBPα) during BMAd differentiation, although no significant changes were observed [8]. All of these findings suggest that sclerostin may play a role in bone-marrow adipogenesis or in the survival of mature BMAds [8,28].

No evidence of an association between circulating sclerostin levels and BMA was found, either at the lumbar spine or proximal femur, which is consistent with the findings in post-menopausal women in the study conducted by Ma et al. [7]. Furthermore, in a single-center cross-sectional pilot study conducted in 15 hemodialysis patients, Wang et al. did not find a correlation between circulating sclerostin and BMA at L3 [29].

Few studies were published on the relationship between circulating sclerostin levels and fat parameters. In a Japanese study on post-menopausal women, the authors reported a positive association between abdominal and gynoid body fat, as assessed using DXA, and sclerostin [27]. More recently, Ma et al. reported that TBF, but not VAT, was associated with circulating sclerostin levels in men, but not in women [7]. Comparing our results with those reported in previous studies is difficult since the parameters used to measure body composition were not always the same as ours. Furthermore, in a mouse model, over-expression of the SOST gene led to an increase in fat mass, whereas knocking out the SOST gene led to a reduction in fat mass [9].

Circulating sclerostin levels correlated negatively with VAT in both groups of post-menopausal women, and with TBF in the cases, but not in the controls. These results are somewhat surprising given that many studies, which are cited above, reported a positive association between sclerostin and fat parameters. However, the results should be interpreted with caution given the correlation coefficients obtained (low or medium) and the exploratory nature of these analyses.

Regarding ALM, previous findings are conflicting. A few studies suggested that lean mass and BMD are interconnected in both older men [30] and post-menopausal women [31]. In a cross-sectional study involving 240 healthy subjects from the Korean Sarcopenic Obesity Study (KSOS), the authors found that circulating sclerostin levels correlated negatively with ALM [10]. Moreover, in a prospective study, Armamento-Vilareal et al. reported that ALM correlated negatively with changes in sclerostin levels in older obese adults [32]. On the other hand, Ma et al. found no association between total lean mass and circulating sclerostin levels, either in men or women [7]. In an animal study, Krause et al. found that SOST knock-out mice exhibited less trabecular bone volume loss and lower lean mass than wild-type mice [33].

Sclerostin levels were correlated negatively with ALM, independently of age, BMD and eGFR in the cases, but not in the controls. Again, these results should be interpreted with caution. Why we failed to find an association between BMA and circulating sclerostin is unclear. Of course, the relationship between fat and bone is complex since associations between adiposity and bone are specific to age, sex, menopausal status, fat deposition and bone compartment [34,35]. Similarly, the evidence shows that the association between BMA and sclerostin is also complex. Circulating sclerostin levels were analyzed, whereas sclerostin is thought to exert its effects primarily at the local level. Although circulating sclerostin levels were shown to correlate strongly with marrow plasma levels in post-menopausal women [36], peripheral levels may not be sensitive enough to detect locally occurring changes in bone. Moreover, serum sclerostin was measured at a single time point only, namely, in fasting morning samples. Since serum sclerostin levels are subject to diurnal variations [36], we cannot rule out the possibility that taking multiple measurements throughout the day, or over several days, would have allowed us to detect a significant association with BMA. Several questions in connection with the measurement of circulating sclerostin remain unanswered, most notably regarding assay performance and various influences on the measurements. For example, marked seasonal variations in serum sclerostin levels were reported, with winter levels being 20% higher than the annual average, and fall levels being 20% lower than the annual average [37].

With regard to body composition, the relationships between sclerostin and fat parameters were quite the opposite of what would be expected. From a theoretical perspective, this finding is counterintuitive given that stem cell lineage allocation is influenced by the Wnt pathway, leading to the hypothesis that higher levels of sclerostin might be associated with an increase in adiposity [7]. However, one explanation for this paradoxical relationship may be that sclerostin is also known to be closely regulated by mechanical forces; serum levels increase in humans after immobilization [38]. Studies also showed that lean mass and fat mass exert mechanical effects on bone [39]. It was reported that osteocytes decrease sclerostin secretion in response to mechanical forces perceived via cellular sensors (integrins, cilium, calcium channels and GPCR) [39].

The negative association found between circulating sclerostin and lean mass was reported in previous studies. Several hypotheses can be put forward to explain this negative relationship, but the accepted explanation is that increased muscle strain acts on osteocyte mechanoreceptors, as described above [39]. Canonical Wnt signaling is one of the major components in skeletal muscle regeneration and myoblast differentiation [39]. Furthermore, Brack et al. reported that the canonical Wnt pathway is associated with fibrogenic changes in aging lean mass and accelerates the aging process [40].

The relationship between sclerostin and body composition parameters is currently difficult to determine due to the lack of available data in humans, including data on the influence of a mutation in the SOST gene on fat mass and lean mass. It was reported that different types of mutations in the SOST gene are responsible for Van Buchem disease and sclerosteosis, both of which are characterized by a considerable increase in bone mass [41]. On this basis, studies sought to investigate the association between single-nucleotide polymorphisms (SNPs) of the SOST gene and body composition in different populations, with conflicting results [42,43]. No clear relationship has yet been established. Furthermore, in these studies, the sample sizes were small and the cohorts were heterogeneous in terms of their origin, sex, age, methods of measuring body composition and SNP selection [43].

The main strength of this study lies in the fact that circulating sclerostin levels and BMA were assessed in a large, well-characterized cohort of post-menopausal women. Other strengths include the fact that many confounders, including BMD, age and eGFR, were used for adjustment. We also chose DXA, which is the clinical standard for BMD measurements. MRI scans, as well as DXA, were all acquired on the same machine. Bone marrow adiposity was measured both at the lumbar spine and hip using MRI, and we took multiple biases into consideration to improve the accuracy [44].

There were, however, several limitations. As this was an observational study, the presence of unmeasured or poorly measured confounders may have influenced our findings. Additionally, our findings apply to post-menopausal women only, and cannot be generalized to men and younger age groups. At this time, assays designed to measure circulating sclerostin levels are still relatively new and have not demonstrated consistent internal agreement. Furthermore, with these assays, differences between serum and plasma levels were observed. Due to the technical uncertainties surrounding the sclerostin assays that are currently available, measurements of circulating sclerostin levels should be interpreted with caution. Finally, our ability to determine whether sclerostin is a mediator in the interactions between bone and BMAT was limited since circulating sclerostin levels were measured, not tissue levels, and circulating sclerostin levels may not fully reflect the tissue levels.

## 4. Materials and Methods

### 4.1. Study Design

This ancillary analysis was an independent research project undertaken with samples and data from the ADIMOS study [45], which is a case–control study involving two groups of post-menopausal women, i.e., a group with fragility fractures that were less than 12 months old, and another group of post-menopausal women with osteoarthritis and no history of fragility fractures. Subjects were recruited from the Department of Rheumatology at Lille University Hospital, France, and were included between October 2018 and June 2021. The parent and ancillary studies were approved by the local Institutional Review Board (2017-A00472-51), and all participants provided their written informed consent. The study procedures complied with the ethical standards of the relevant institutional and national Human Experimentation Ethics Committees.

### 4.2. Study Population

For the cases, the inclusion criteria were (i) post-menopausal women between 50 and 90 years old, (ii) living in France and (iii) seen by the Fracture Liaison Service at Lille University Hospital for fragility fractures due to low-energy trauma (e.g., a fall from standing height). Fragility fractures were hip fractures, vertebral fractures, proximal humerus fractures, pelvis fractures, ribs fractures, and distal forearm or wrist fractures. To be eligible for the study, patients must have been included and interviewed within 12 months of diagnosis of the fracture event.

For the controls, the inclusion criteria were (i) post-menopausal women between 50 and 90 years old, (ii) living in France and (iii) seen by the Department of Rheumatology at Lille University Hospital for osteoarthritis (hips, knees, hands or spine). Controls were eligible for the analysis if they reported no previous history of fragility fractures after the age of 40.

Exclusion criteria were (i) contraindication for magnetic resonance imaging (MRI); (ii) body mass index (BMI) > 38 kg/m^2^; (iii) weight > 140 kg; (iv) chronic kidney disease with calculated creatinine clearance < 30 mL/min; (v) a disease known to affect bone metabolism; (vi) current use of medications known to affect BMD, including oral glucocorticoids; (vii) treatments for osteoporosis; and (viii) menopausal hormone therapy (MHT). Prior use of treatment for osteoporosis and MHT over 12 months old was allowed. The exclusion criteria were the same for both cases and controls.

For the ancillary study, eligible participants must have participated in the ADIMOS study and have had a sufficient number of aliquots in the Lille University Hospital’s serum bank.

### 4.3. Study Protocol

This ancillary analysis utilized data obtained for the parent study over the course of a visit day where participants were invited to undergo (i) an MRI scan of the lumbar spine (L1 to L4) and non-dominant hip for BMA quantification, (ii) a DXA scan to assess their lumbar spine and hip BMD, and (iii) body composition and blood tests. Data on medical history, treatments, osteoporosis risk factors, calcium intake, physical activity and anthropometric features were collected during a medical visit. The study protocol is presented in detail below.

### 4.4. Disease Assessment

Demographic and clinical characteristics were recorded by one physician. Data on risk factors for osteoporosis were collected and included current smoking, excessive alcohol consumption, previous use of oral corticosteroids (exposed to ≥5 mg/day of prednisolone for ≥3 months), history of fragility fractures after the age of 40 and family history of osteoporosis (hip fracture in mother or father). Data on prior use of MHT and treatments for osteoporosis over 12 months old were also collected. Other data, such as the Charlson Comorbidity Index (CCI), leisure time activity (score 0–15) and medication were collected for all patients.

### 4.5. Bone Marrow Adiposity Measurement Using MRI

#### 4.5.1. Image Acquisition

All subjects underwent an MRI examination on a 3.0 T scanner (Ingenia; Philips Healthcare, Best, The Netherlands) under the supervision of a senior musculoskeletal radiologist. Imaging was performed using a conventional protocol that included T1- and T2-weighted 2-point Dixon turbo-spin echo (TSE) acquisitions in the sagittal plane. Immediately after this clinical exploration, the BMA was quantified using a six-echo three-dimensional gradient-echo sequence (mDixon-Quant; Philips Healthcare, Best, The Netherlands), permitting chemical-shift encoding-based water–fat separation at the lumbar spine (sagittal) and the non-dominant hip (coronal oblique). Offline reconstructions of computed proton density fat fraction maps (PDFF; ratio of the fat signal over the fat and water signals) were created using a precalibrated seven-peak fat spectrum and a single T2* correction.

#### 4.5.2. MR Segmentation

The MRI acquisitions obtained for each subject were reviewed by a senior musculoskeletal radiologist on a dedicated workstation using IntelliSpace Portal (Philips Healthcare; Best, The Netherlands) for segmentation. A morphological assessment was first performed to search for transitional anomalies, severe degenerative changes, and bone-marrow-replacing lesions at the hip or lumbar spine.

Subsequently, the three most central lumbar spine slices were chosen based on the PDFF maps computed from the mDixon-Quant acquisitions. A polygonal region of interest (ROI) was drawn around the L1–L4 vertebral body, avoiding fractured vertebrae, the immediate subchondral bone, bone-marrow-replacing lesions, severe degenerative changes and the basivertebral vein. Similarly, an ROI was drawn around the femoral neck and the femoral diaphysis based on the three most central coronal oblique slices from the mDixon-Quant acquisitions of the non-dominant hip.

### 4.6. Bone Mineral Density and Body Composition Assessment

Bone mineral density was measured using DXA (HOLOGIC Discovery A S/N 81360, Rheumatology Department). The machine was calibrated daily, and quality assurance tests were carried out daily and weekly. Data included the BMD (in g/cm^2^ of hydroxyapatite) at three sites: lumbar spine, total hip and femoral neck. As recommended by WHO, osteoporosis was defined as T-score ≤ −2.5.

Whole-body dual-energy X-ray absorptiometry (DXA) scans were performed to evaluate the body composition. Data included appendicular lean mass (ALM, kg), TBF (kg) and visceral adipose tissue (VAT, cm^2^). Appendicular lean mass was defined as the sum of the lean soft tissue masses of the four limbs.

### 4.7. Laboratory Measurements

#### 4.7.1. Samples

All blood samples were obtained from peripheral blood drawn early in the morning on the day of admission to the hospital. The samples were then centrifuged, and the serum was portioned, frozen and stored as aliquots at −80 °C at the Lille University Hospital’s Biological Resource Center until used for the measurement of sclerostin (ng/mL), adiponectin (ng/mL) and leptin (ng/mL).

Total calcium, creatinine and high-sensitivity C-reactive protein (hs-CRP) were assessed using routine assays. The estimated glomerular filtration rate (eGFR) was calculated using the CKD-EPI formula (mL/min). Intact parathormone (iPTH) was measured using a chemiluminescent immunoassay on an Architect automatic analyzer (Abbott Laboratories, Abbott Park, IL, USA). 25-Hydroxyvitamin D (25(OH) D) was measured using a competitive chemiluminescent immunoassay on an IDSiSYS instrument (IDS, Pouilly-en-Auxois, France).

#### 4.7.2. Sclerostin

ELISA kits were purchased from TECO Medical, Sissach, Switzerland (Human Sclerostin EIA, High Sensitivity). Serum samples were analyzed and sclerostin concentrations were measured following the manufacturer’s instructions. The TECO Medical high-sensitivity kit plates were washed with 350 µL wash buffer (provided) for 2 min at room temperature, then blot dried. The wells were then loaded with 25 µL standards, controls and samples, followed by 50 μL matrix and 50 μL antibody solutions. The plates were sealed and incubated on a shaker at 500 rpm for 4 h. The wells were washed four times with 350 µL wash buffer and then developed in the dark with 100 μL of an antigen–antibody complex solution, namely, 3,3′,5,5′-tetramethylbenzidine (TMB), at room temperature for 30 min. The reaction was stopped with 100 μL stop solution. Absorbance was measured at 450 nm, with a reference filter between 590 and 650 nm. Samples were run in duplicate, with an average coefficient of variation between runs of 2.5% for sclerostin. Concentrations with high CV percentages (>10%) were excluded from the analysis.

#### 4.7.3. Leptin and Adiponectin

Serum adiponectin and leptin levels were measured using the human Quantikine ELISA Kit (R&D Systems, Minneapolis, MN, USA) in line with the manufacturer’s instructions. In the beginning, all tested samples of serum were diluted 100 times with assay diluent according to the manufacturer’s instructions.

Standard or diluted serum samples were added to coated microplates and incubated for 3 h at room temperature. The plates were then washed and incubated with a premixed capture and detection antibody cocktail for 1 h at room temperature. After washing, a 3,3′,5,5′-tetramethylbenzidine (TMB) substrate was added to each well and incubated for 30 min to detect the reaction of antigen–antibody complexes. Finally, to terminate the peroxidase reaction, a stop solution was added to each well, and the optical density (OD) was measured at 450 nm. All quantitations were performed in duplicate, and the mean value of the 2 measurements was used as the result. The average intra-assay CVs of leptin and adiponectin were 2.7% and 3.1%, respectively. Concentrations with low CV percentages (>10%) were excluded from the analysis.

### 4.8. Statistical Analysis

Categorical variables were expressed as number (percentage). Quantitative variables were expressed as mean (standard deviation (SD)), or as median (interquartile range (IQR)) for non-Gaussian distributions. The normality of distributions was assessed using histograms and the Shapiro–Wilk test.

Comparisons of patient characteristics, biochemistry, comorbidities, risk factors, body composition, BMD and PDFF measurements between the cases and controls were performed using Student’s *t*-test for quantitative variables (or the Mann–Whitney U test for non-Gaussian distributions), and the chi-squared test was used for categorical variables (or Fisher’s exact test when the expected cell frequency was <5).

The correlation between sclerostin and the parameters of interest in the case and control groups was assessed separately by calculating Pearson’s correlation coefficients, or Spearman’s rank correlation coefficients for non-Gaussian distributions, with their 95% confidence intervals.

Comparisons of the primary objectives (circulating sclerostin levels and PDFF at the lumbar spine, femoral neck and femoral diaphysis) and the secondary objectives (circulating sclerostin levels and body composition parameters, including TBF (kg), VAT (cm^2^) and ALM (kg)) between cases and controls were performed using several models: model 1—without adjustment; model 2—after adjusting for age and eGFR; and model 3—after adjusting for age, eGFR and BMD. Partial correlations were used in models in which adjustments were made.

Correlation coefficients (r) in absolute values of 0.1 to 0.3, 0.3 to 0.5 and 0.5 to 1.0 were interpreted as small, medium and large correlations, respectively.

As this was an exploratory analysis, no adjustments for multiple testing were made. As such, secondary outcomes and correlation analyses should be interpreted with caution and as hypothesis-generating.

Statistical testing was conducted at a two-tailed α-level of 0.05. Data were analyzed using SAS software version 9.4 (SAS Institute, Cary, NC, USA).

## 5. Conclusions

In this study on post-menopausal women with and without fragility fractures, we found (i) no relationship between circulating sclerostin levels and BMA, and (ii) a negative correlation between circulating sclerostin levels and body composition parameters, such as VAT, TBF and ALM. As anti-sclerostin antibodies (e.g., romosozumab) were shown to be effective in the treatment of post-menopausal osteoporosis, data on their effects on BMA and body composition are eagerly awaited.

## Figures and Tables

**Table 1 ijms-24-05922-t001:** Patients’ general characteristics and biochemistry results at baseline.

	N	Cases (*n* = 100)	N	Controls (*n* = 99)	*p*-Value
Age (years)	100	70.2 ± 10.6	99	64.7 ± 8.5	<0.001
Weight (kg)	100	67.2 ± 15.6	99	72.1 ± 15.8	0.03
Height (cm)	100	159.1 ± 6.8	99	161.2 ± 6.2	0.02
BMI (kg/m^2^)	100	26.5 ± 5.9	99	27.7 ± 5.8	0.14
Leisure time activity (score 0–15)	100	8.7 ± 2.6	99	9.3 ± 2.4	0.07
**Comorbidities**					
Type 2 diabetes	100	12 (12.0)	99	12 (12.1)	0.98
Charlson Comorbidity Index	100	3 (2 to 5)	99	2 (0 to 4)	<0.001
**Osteoporosis Risk Factors**					
Excessive alcohol consumption	100	8 (8.0)	99	4 (4.0)	0.24
Current smoking	100	13 (13.0)	99	10 (10.1)	0.52
Family history of hip fracture	100	11 (11.0)	99	11 (11.1)	0.98
Previous use of corticosteroids	100	6 (6.0)	99	4 (4.0)	0.75
**Biochemistry Results**					
Sclerostin (ng/mL)	99 ^1^	0.7 ± 0.2	99	0.7 ± 0.2	0.60
Leptin (ng/mL)	99 ^1^	18.6 (9.7 to 31.4)	99	23.7 (13.2 to 37.6)	0.11
Adiponectin (μg/mL)	95 ^2^	9.2 (4.9 to 15.3)	98 ^3^	7.8 (5.00 to 12.6)	0.10
Calcium (mmol/L)	100	2.4 ± 0.1	99	2.4 ± 0.1	0.33
25(OH) vitamin D (ng/mL)	100	30.1 ± 12.8	99	26.4 ± 9.9	0.03
Serum PTH (pg/mL)	100	42.0 (30.0 to 56.5)	99	47.0 (38.0 to 59.0)	0.05
Creatinine (µmol/L)	100	62.0 (53.0 to 71.0)	99	62.0 (62.0 to 71.0)	0.95
Creatinine clearance (MDRD formula) (mL/min)	100	82.1 (67.1 to 94.0)	99	88.0 (77.0 to 95.5)	0.06
**Bone Mineral Density**					
Lumbar spine BMD (g/cm^2^)	99 ^4^	0.847 ± 0.169	99	0.939 ± 0.174	<0.001
Total hip BMD (g/cm^2^)	97 ^5^	0.757 ± 0.135	99	0.866 ± 0.145	<0.001
Femoral neck BMD (g/cm^2^)	97 ^5^	0.632 ± 0.127	99	0.726 ± 0.122	<0.001
**Bone Marrow Adiposity (mDixon-Quant)**					
Lumbar spine PDFF (%)	100	59.1 ± 9.6	99	56.6 ± 9.4	0.06
Femoral diaphysis PDFF (%)	95 ^6^	81.4 ± 8.5	97 ^7^	79.8 ± 9.8	0.23
Femoral neck PDFF (%)	95 ^6^	82.2 ± 8.1	97 ^7^	81.5 ± 8.5	0.56
**Body Composition**					
TBF (kg)	97 ^8^	29.8 ± 10.6	98 ^9^	32.3 ± 11.1	0.37
VAT (cm^2^)	97 ^8^	146 ± 72.2	98 ^9^	164.5 ± 89.3	0.12
ALM (kg)	97 ^8^	13.1 ± 2.9	98 ^9^	13.8 ± 2.33	0.07

Values expressed as numbers (%), mean ± SD or median (IQR). Abbreviations: SD—standard deviation; IQR—interquartile range; PTH—parathyroid hormone; TBF—total body fat; VAT—visceral adiposity; ALM—appendicular lean mass; PDFF—MR-based proton density fat fraction. ^1^ Serum sclerostin and leptin measurements were not included for 2 patients (unacceptable quantity of serum). ^2^ Serum adiponectin measurements were not included for 4 patients (unacceptable quantity of serum, *n* = 1; high coefficient of variation percentage (>10%), *n* = 1; too concentrated despite 100-fold dilution, *n* = 2). ^3^ Serum adiponectin measurements were not included for 1 patient (high coefficient of variation percentage). ^4^ Lumbar spine BMD measurements were not included for 1 woman (vertebral fractures at L1, L2 and L3). ^5^ Hip BMD measurements were not included for 3 women (bilateral hip arthroplasty). ^6^ Hip PDFF measurements were not included for 5 women (bilateral hip arthroplasty, *n* = 3; unacceptable quality of measurements, *n* = 2). ^7^ Hip PDFF measurements were not included for 2 women (bilateral hip osteonecrosis, *n* = 1; unacceptable quality of measurements, *n* = 1). ^8^ Body composition measurements were not included for 3 women (unacceptable quality of measurements). ^9^ Body composition measurements were not included for 1 woman (unacceptable quality of measurements).

**Table 2 ijms-24-05922-t002:** Correlations between circulating sclerostin and parameters of interest.

	*N*	Cases, *n* = 99	*N*	Controls, *n* = 99
		Sclerostin (ng/mL)		Sclerostin (ng/mL)
Age (years)	99	R = 0.09 (−0.11 to 0.29) *p* = 0.35	99	R = 0.17 (−0.03 to 0.35) *p* = 0.09
Lumbar spine BMD (g/cm^2^)	98 ^1^	R = 0.42 (0.24 to −0.57) *p* < 0.001	99	R = 0.56 (0.41 to 0.68) *p* < 0.001
Femoral neck BMD (g/cm^2^)	96 ^2^	R = 0.43 (0.25 to 0.58) *p* < 0.001	99	R = 0.27 (0.07 to 0.44) *p* = 0.007
Total hip BMD (g/cm^2^)	96 ^2^	R = 0.40 (0.22 to 0.56) *p* < 0.001	99	R = 0.35 (0.17 to 0.51) *p* < 0.001
Leptin (ng/mL)	99	R = −0.15 (−0.34 to 0.05) *p* = 0.15	99	R = 0.12 (−0.07 to 0.31) *p* = 0.22
Adiponectin (μg/mL)	95 ^3^	R = 0.08 (−0.13 to 0.28) *p* = 0.45	98 ^4^	R = −0.02 (−0.22 to 0.17) *p* = 0.83
Creatinine clearance (CKD-EPI formula) (mL/min)	99	R = −0.22 (−0.40 to −0.02) *p* = 0.03	99	R = −0.29 (−0.46 to −0.10) *p* = 0.003
Serum PTH (pg/mL)	99	R = −0.23 (−0.41 to −0.04) *p* = 0.02	99	R = −0.09 (−0.28 to 0.11) *p* = 0.36

Abbreviations: BMD—bone mineral density; PTH—parathyroid hormone. ^1^ Lumbar spine BMD measurements were not included for 1 woman (vertebral fractures at L1, L2 and L3). ^2^ Hip BMD measurements were not included for 3 women (bilateral hip arthroplasty). ^3^ Serum adiponectin measurements were not included for 4 women (unacceptable quantity of serum, *n* = 1; high coefficient of variation percentage (>10%), *n* = 1; too concentrated despite 100-fold dilution, *n* = 2). ^4^ Serum adiponectin measurements were not included for 1 patient (high coefficient of variation percentage).

**Table 3 ijms-24-05922-t003:** Relationship between SOST and lumbar spine, femoral neck and femoral diaphysis PDFF.

		Model 1		Model 2		Model 3	
	N	R	*p*	R	*p*	R	*p*
Cases (*n* = 99)							
Lumbar spine PDFF	99	R = 0.008 (0.19 to 0.21)	*p* = 0.93	R = −0.03 (−0.23 to 0.17)	*p* = 0.75	R = 0.001 (−0.20 to 0.20)	*p* = 0.99
Femoral neck PDFF	95 ^1^	R = −0.002 (−0.20 to 0.20)	*p* = 0.98	R = −0.001 (−0.20 to 0.20)	*p* = 1.00	R = 0.05 (−0.16 to 0.25)	*p* = 0.65
Femoral diaphysis PDFF	95 ^1^	R = −0.04 (−0.24 to 0.16)	*p* = 0.67	R = −0.04 (−0.24 to 0.16)	*p* = 0.68	R = 0.09 (−0.12 to 0.29)	*p* = 0.41
Controls (*n* = 99)							
Lumbar spine PDFF	99	R = 0.04 (−0.16 to 0.24)	*p* = 0.70	R = −0.02 (−0.22 to 0.18)	*p* = 0.84	R = 0.11 (−0.10 to 0.30)	*p* = 0.30
Femoral neck PDFF	97 ^2^	R = −0.17 (−0.36 to 0.03)	*p* = 0.09	R = −0.17 (−0.36 to 0.03)	*p* = 0.09	R = −0.10 (−0.30 to 0.10)	*p* = 0.32
Femoral diaphysis PDFF	97 ^2^	R = −0.17 (−0.36 to 0.03)	*p* = 0.09	R = −0.16 (−0.35 to 0.04)	*p* = 0.11	R = −0.008 (−0.21 to 0.19)	*p* = 0.94

Abbreviations: PDFF—proton density fat fraction. Model 1: without adjustment. Model 2: *p*-values were adjusted for age and eGFR. Model 3: *p*-values were adjusted for age, eGFR and bone mineral density. ^1^ Hip PDFF measurements were not included for 4 women (bilateral hip arthroplasty, *n* = 3; unacceptable quality of measurements, *n* = 1). ^2^ Hip PDFF measurements were not included for 2 women (bilateral hip osteonecrosis, *n* = 1; unacceptable quality of measurements, *n* = 1).

**Table 4 ijms-24-05922-t004:** Relationship between SOST and visceral adipose tissue, total body fat and appendicular lean mass.

		Model 1	Model 2			Model 3	
	N	R	*p*	R	*p*	R	*p*
			**Sclerostin (ng/mL)**				
Cases (*n* = 99)							
Total body fat (kg)	95 ^1^	R = −0.18 (−0.37 to 0.02)	*p* = 0.07	R = −0.21 (−0.40 to −0.01)	*p* = 0.04	R = −0.47 (−0.61 to −0.29)	*p* < 0.001
VAT (cm^2^)	95 ^1^	R = −0.14 (−0.33 to 0.07)	*p* = 0.19	R = −0.18 (−0.37 to 0.02)	*p* = 0.08	R = −0.32 (−0.49 to −0.12)	*p* = 0.002
ALM (kg)	95 ^1^	R = −0.04 (−0.24 to −0.16)	*p* = 0.70	R = −0.11 (−0.31 to 0.09)	*p* = 0.29	R = −0.26 (−0.44 to −0.06)	*p* = 0.01
Controls (*n* = 99)							
Total body fat (kg)	98 ^2^	R = 0.15 (−0.05 to 0.34)	*p* = 0.14	R = 0.10 (−0.10 to 0.29)	*p* = 0.34	R = −0.13 (−0.32 to 0.08)	*p* = 0.21
VAT (cm^2^)	98 ^2^	R = 0.14 (−0.06 to 0.32)	*p* = 0.18	R = 0.05 (−0.15 to 0.25)	*p* = 0.60	R = −0.24 (−0.42 to −0.04)	*p* = 0.02
ALM (kg)	98 ^2^	R = 0.25 (0.05 to 0.43)	*p* = 0.01	R = 0.22 (0.02 to 0.40)	*p* = 0.03	R = 0.06 (−0.14 to 0.25)	*p* = 0.56

Abbreviations: TBF—total body fat; VAT—visceral adiposity; ALM—appendicular lean mass; PDFF—MR-based proton density fat fraction. Model 1: without adjustment. Model 2: *p*-values were adjusted for age and eGFR. Model 3: *p*-values were adjusted for age, eGFR and lumbar spine bone mineral density. ^1^ Body composition measurements were not included for 3 women (unacceptable quality of measurements). ^2^ Body composition measurements were not included for 1 woman (unacceptable quality of measurements).

## Data Availability

The data that support the findings of this study are available on request from the corresponding author, Paccou J.

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
