# Peer review of "Relationships between Circulating Sclerostin, Bone Marrow Adiposity, Other Adipose Deposits and Lean Mass in Post-Menopausal Women"

_ijms, 2023, doi:10.3390/ijms24065922_

Round 1

Reviewer 1 Report

An interesting study is presented in this manuscript examining the relationships between circulating sclerostin levels, bone mar[1]row adiposity, other adipose deposits, and lean mass in post[1]menopausal women.

The outcome form the study the authors concluded that no evidence of a relationship between serum sclerostin and BMA was found. Additionally, they also found with there  in vitro experiments, we found no evidence that hSOST influences BMAds

In the MRI segmentation analysis “Similarly, a ROI was drawn around the femoral neck and the femoral diaphysis, based on the three most central coronal oblique slices from the mDixon-Quant acquisitions of the non-dominant hip”  Was this analysis blinded? Similarly with BMD and body composition analysis

What were the seeding densities of the cell culture expts?

In fig 1 replace J with D – days?

Fig 2 is of inadequate quality. Need to add scale bars

Is the differentiation of 7 days long enough for Adipogenesis generation for the studies run?

Author Response

Response to Editor and Reviewers

Manuscript ID: ijms-2234184

Relationships between circulating sclerostin levels, bone marrow adiposity, other adipose deposits, and lean mass in postmenopausal women

Dear Editor,

Thank you for giving us the opportunity to submit a revised version of our article entitled “Relationships between circulating sclerostin levels, bone marrow adiposity, other adipose deposits, and lean mass in postmenopausal women”. We very much appreciate the reviewers’ detailed comments and would like to thank them for their evaluation of our manuscript and their constructive suggestions. We are pleased to report that we have been able to address all of them fully, which led to a substantial improvement of our manuscript. Please find our responses below.

Professor Julien Paccou, M.D., Ph.D.

Department of Rheumatology, Lille University Hospital.

Phone: + 33 3 22 82 77 90 Fax: + 33 3 22 82 74 69.

[email protected]

Reviewer #1: An interesting study is presented in this manuscript examining the relationships between circulating sclerostin levels, bone marrow adiposity, other adipose deposits, and lean mass in postmenopausal women.

The outcome form the study the authors concluded that no evidence of a relationship between serum sclerostin and BMA was found. Additionally, they also found with there in vitro experiments, we found no evidence that hSOST influences BMAds.

1- In the MRI segmentation analysis “Similarly, a ROI was drawn around the femoral neck and the femoral diaphysis, based on the three most central coronal oblique slices from the mDixon-Quant acquisitions of the non-dominant hip” Was this analysis blinded? Similarly with BMD and body composition analysis

Author’s reply: Thank you for this relevant comment. The post-processing of mDixon-Quant and MRS acquisitions was performed based on a patient list. However, the hip contouring was not blinded from the spine analysis. Subsequently, a vertebral fracture previously observed on a spine acquisition highly suggested the participant’s group. In other words, the analysis could not be purely blinded. Nevertheless, choosing the three exact most central slices limited subjective biases. The excellent intra- and inter-observer agreement (>0.90 for both spine and hip) confirmed this methodology.

Regarding BMD and body composition analysis, the semi-automatic post-processing was performed regardless of the provided medical history. Data were consequently obtained blindly from the subject’s group.

2- What were the seeding densities of the cell culture expts?

Author’s reply: As requested by Reviewer 3, we have deleted the part on basic research.

3- In fig 1 replace J with D – days?

Author’s reply: As requested by Reviewer 3, we have deleted the part on basic research.

4- Fig 2 is of inadequate quality. Need to add scale bars

Author’s reply: As requested by Reviewer 3, we have deleted the part on basic research.

5- Is the differentiation of 7 days long enough for Adipogenesis generation for the studies run?

Author’s reply: As requested by Reviewer 3, we have deleted the part on basic research.

Reviewer 2 Report

Sclerostin apparently increases bone marrow adipocytes by a previously reported process of Wnt signalling and reduced bone formation.  Various aspects of the biochemically complex background are subject to further investigation concerning the effect of circulating sclerostin on bone marrow adipocytes in postmenopausal women with and without fractures and the results have current clinical relevance to vanguard romosozumab treatment.

Title: Informative, would be more concise by removing “levels”.

Abstract:  Aims to compare the relationship of sclerostin and bone marrow adipose tissue variables between fracture and nonfracture postmenopausal women, with reference also to body composition (MRI, DXA) and an in vitro analysis of sclerostin on adipocytes.

Introduction: Background presentation of sclerostin as a glycoprotein of diurnal significance secreted by osteocytes, and circulating in levels correlating with age, sex, BMD and renal function.

Methods: Objectives are described in detail using established technology with extensive statistical analysis and with the benefit of a well organised Figure 1.

Results: General characteristics and biochemistry in fracture and nonfracture groups are presented in a clear Table 1 indicating parameters of interest and sclerostin correlation status with various factors (including BMD, age, glomerular filtration rate, leptin, PTH), summarised in well-constructed Tables 2,3 and 4. In addition is gene expression in cell culture when fibroblast cells become rounded with lipid droplets in seven days. Observations summarised in Figures 2 and 3.

Discussion:  An unexpected observation was the apparent absence of a correlation between circulating sclerostin and bone marrow adipocytes. Also, in contrast to other reports, sclerostin did not correlate with age in postmenopausal women. The authors await the outcome by others of ongoing romsozumab anti-sclerostin treatment on postmenopausal osteoporosis and bone marrow adipocytes (FRAME study of transiliac bone biopsies). Other current factors of interest include a seasonal variation with 20% higher sclerostin in winter, together with the intervention of mechanical forces whereby sclerostin rises with immobilisation and osteocytes produce sclerostin in response to perceived biomechanical sensors (integrins, cilium etc). Finally, the authors express a reservation that circulating sclerostin may not fully reflect tissue levels.  

References: The fact that these are all recent seems to illustrate the present “cutting edge” status of the subject. The addition of a historical context to the fundamental histology would be helpful, including courteous reference to pioneer authors such as Pierre Meunier of Lyons (Osteoporosis and the replacement of cell populations of the marrow by adipose tissue. Clin Orthop 80, 147-154, 1971.)

Editorial comments: As English is not the first language of the authors the following comments might be considered. A more concise, less personal style is preferable; throughout there is more emphasis than necessary on “we” and “our” and there is too much repetition of “in this study” and “in our study.” There is no place in science for “we believe “ (i.e., “the evidence shows” is more authoritative). Also “to trend towards” lacks style, as also does “explored” (“examined” is more appropriate). Finally, the overuse of “study” becomes tedious (replace occasionally with “investigation” or “analysis”.

The list of abbreviations is especially useful for non-biochemists as they can be a distraction from the text.

Author Response

Response to Editor and Reviewers

Manuscript ID: ijms-2234184

Relationships between circulating sclerostin levels, bone marrow adiposity, other adipose deposits, and lean mass in postmenopausal women

Dear Editor,

Thank you for giving us the opportunity to submit a revised version of our article entitled “Relationships between circulating sclerostin levels, bone marrow adiposity, other adipose deposits, and lean mass in postmenopausal women”. We very much appreciate the reviewers’ detailed comments and would like to thank them for their evaluation of our manuscript and their constructive suggestions. We are pleased to report that we have been able to address all of them fully, which led to a substantial improvement of our manuscript. Please find our responses below.

Professor Julien Paccou, M.D., Ph.D.

Department of Rheumatology, Lille University Hospital.

Phone: + 33 3 22 82 77 90 Fax: + 33 3 22 82 74 69.

[email protected]

Reviewer #2:

1- Sclerostin apparently increases bone marrow adipocytes by a previously reported process of Wnt signaling and reduced bone formation.  Various aspects of the biochemically complex background are subject to further investigation concerning the effect of circulating sclerostin on bone marrow adipocytes in postmenopausal women with and without fractures and the results have current clinical relevance to vanguard romosozumab treatment.

Author’s reply: Thank you for this comment.

2- Title: Informative, would be more concise by removing “levels”.

Author’s reply: we have edited the title.

3- Abstract:  Aims to compare the relationship of sclerostin and bone marrow adipose tissue variables between fracture and nonfracture postmenopausal women, with reference also to body composition (MRI, DXA) and an in vitro analysis of sclerostin on adipocytes.

Author’s reply: Thank you for this comment.

4- Introduction: Background presentation of sclerostin as a glycoprotein of diurnal significance secreted by osteocytes, and circulating in levels correlating with age, sex, BMD and renal function.

Author’s reply: Thank you for this comment.

5- Methods: Objectives are described in detail using established technology with extensive statistical analysis and with the benefit of a well organised Figure 1.

Author’s reply: Thank you for this comment. As requested by Reviewer 2, we have deleted the part on basic research including Figure 1.

6- Results: General characteristics and biochemistry in fracture and nonfracture groups are presented in a clear Table 1 indicating parameters of interest and sclerostin correlation status with various factors (including BMD, age, glomerular filtration rate, leptin, PTH), summarised in well-constructed Tables 2,3 and 4. In addition is gene expression in cell culture when fibroblast cells become rounded with lipid droplets in seven days. Observations summarised in Figures 2 and 3.

Author’s reply: Thank you for these comments. As requested by Reviewer 3, we have deleted the part on basic research including Figures 2 and 3.

7- Discussion:  An unexpected observation was the apparent absence of a correlation between circulating sclerostin and bone marrow adipocytes. Also, in contrast to other reports, sclerostin did not correlate with age in postmenopausal women. The authors await the outcome by others of ongoing romsozumab anti-sclerostin treatment on postmenopausal osteoporosis and bone marrow adipocytes (FRAME study of transiliac bone biopsies). Other current factors of interest include a seasonal variation with 20% higher sclerostin in winter, together with the intervention of mechanical forces whereby sclerostin rises with immobilisation and osteocytes produce sclerostin in response to perceived biomechanical sensors (integrins, cilium etc). Finally, the authors express a reservation that circulating sclerostin may not fully reflect tissue levels.

Author’s reply: Thank you for these comments.

8- References: The fact that these are all recent seems to illustrate the present “cutting edge” status of the subject. The addition of a historical context to the fundamental histology would be helpful, including courteous reference to pioneer authors such as Pierre Meunier of Lyons (Osteoporosis and the replacement of cell populations of the marrow by adipose tissue. Clin Orthop 80, 147-154, 1971.)

Author’s reply: we have edited the manuscript and added some references including Meunier P et al.

Meunier P, Aaron J, Edouard C, Vignon G. Osteoporosis and the replacement of cell populations of the marrow by adipose tissue. A quantitative study of 84 iliac bone biopsies. Clin Orthop Relat Res. 1971 Oct;80:147-54

9- Editorial comments: As English is not the first language of the authors the following comments might be considered. A more concise, less personal style is preferable; throughout there is more emphasis than necessary on “we” and “our” and there is too much repetition of “in this study” and “in our study.” There is no place in science for “we believe “ (i.e., “the evidence shows” is more authoritative). Also “to trend towards” lacks style, as also does “explored” (“examined” is more appropriate). Finally, the overuse of “study” becomes tedious (replace occasionally with “investigation” or “analysis”.

Author’s reply: we have edited the manuscript.

10- The list of abbreviations is especially useful for non-biochemists as they can be a distraction from the text.

Author’s reply: Thank you for these comments.

Reviewer 3 Report

Major comment

The authors divided the manuscript into two parts: one clinical and one basic research. Since the basic research part is small and more experiments would be needed to prove what the authors want, I suggest eliminating this part and focusing only on the clinical part. Later the authors could publish the basic research part with more experiments in order to demonstrate irrefutably that adding hSOST has no effect on adipogenic cell differentiation.

Other comments:

“Bone marrow-derived stromal cell (BMSC) allocation is influenced by the Wnt path-way, leading to the hypothesis that higher levels of sclerostin might be associated with an increase in bone marrow adiposity (BMA)”. “BMSC allocation” is not clear.

“In the Iceland AGES-Reykjavik cohort study, the authors reported a positive association between circulating sclerostin levels and total body fat (TBF) in men, but not in women (7).” This part is repeated twice in the introduction. The authors should better organize the introduction avoiding repetitions.

It is not clear to me why did the authors add BSA to study the influence of sclerostin in vitro.

“objectives” section should be deleted from the methods. Objectives/aims of the study should be defined at the end of the introduction.

Authors should use the same names to define the two groups in the results. For example, in table 1 fracture vs no fractures is used, while in table 2 controls vs cases.

Figure 2 is not clear. Images with higher quality should be added. Scale bar/magnification should  be added.

The authors reported “Moreover, the expression levels of the adipogenic markers PPARγ2, C/EBPα (two key transcription factor of adipocyte differentiation) and Plin1 (a marker associated with lipid droplet formation) in adipocytes was measured after 7 days of differenti-ation, and the gene expression of these markers was found to be higher in BMAds than in hBMSCs.” However, these results are not reported. They should be added.

Reporting only the gene expression of three markers is not enough to demonstrate that adding hSOST has no impact on the adipogenic differentiation. Other markers should be added, including protein quantification.

Subsections should be deleted from the discussion.

The conclusions should be improved reporting not only what the authors did not find but also the results found.

“In the FRAME study, transiliac bone biopsies were performed at 2 or 12 months of treatment with romosozumab or placebo to assess BMAT parameters (43). We are look-ing forward to the results, which are expected to be published shortly.” This part should be deleted from the conclusion as it is not clear.

Line numbers should be added.

The authors are encouraged to follow the guidelines of the journal. References should be checked and reported between [] and not (). Section and subsections should be numbered. Table should be revised following the layout of the journal.

Author Response

Response to Editor and Reviewers

Manuscript ID: ijms-2234184

Relationships between circulating sclerostin levels, bone marrow adiposity, other adipose deposits, and lean mass in postmenopausal women

Dear Editor,

Thank you for giving us the opportunity to submit a revised version of our article entitled “Relationships between circulating sclerostin levels, bone marrow adiposity, other adipose deposits, and lean mass in postmenopausal women”. We very much appreciate the reviewers’ detailed comments and would like to thank them for their evaluation of our manuscript and their constructive suggestions. We are pleased to report that we have been able to address all of them fully, which led to a substantial improvement of our manuscript. Please find our responses below.

Professor Julien Paccou, M.D., Ph.D.

Department of Rheumatology, Lille University Hospital.

Phone: + 33 3 22 82 77 90 Fax: + 33 3 22 82 74 69.

[email protected]

Reviewer #3:

Major comment

The authors divided the manuscript into two parts: one clinical and one basic research. Since the basic research part is small and more experiments would be needed to prove what the authors want, I suggest eliminating this part and focusing only on the clinical part. Later the authors could publish the basic research part with more experiments in order to demonstrate irrefutably that adding hSOST has no effect on adipogenic cell differentiation.

Author’s reply: As requested by Reviewer 3, we have deleted the part on basic research.

Other comments:

1-“Bone marrow-derived stromal cell (BMSC) allocation is influenced by the Wnt path-way, leading to the hypothesis that higher levels of sclerostin might be associated with an increase in bone marrow adiposity (BMA)”. “BMSC allocation” is not clear.

Author’s reply: we have changed “BMSC allocation” by “BMSC differentiation”.

2-“In the Iceland AGES-Reykjavik cohort study, the authors reported a positive association between circulating sclerostin levels and total body fat (TBF) in men, but not in women (7).” This part is repeated twice in the introduction. The authors should better organize the introduction avoiding repetitions.

Author’s reply: we have edited the introduction.

3- It is not clear to me why did the authors add BSA to study the influence of sclerostin in vitro.

Author’s reply: As requested by Reviewer 3, we have deleted the part on basic research.

4- “Objectives” section should be deleted from the methods. Objectives/aims of the study should be defined at the end of the introduction.

Author’s reply: we have edited the manuscript.

5- Authors should use the same names to define the two groups in the results. For example, in table 1 fracture vs no fractures is used, while in table 2 controls vs cases.

Author’s reply: we have edited the manuscript.

6- Figure 2 is not clear. Images with higher quality should be added. Scale bar/magnification should be added.

Author’s reply: As requested by Reviewer 3, we have deleted the part on basic research including Figure 2.

7- The authors reported “Moreover, the expression levels of the adipogenic markers PPARγ2, C/EBPα (two key transcription factor of adipocyte differentiation) and Plin1 (a marker associated with lipid droplet formation) in adipocytes was measured after 7 days of differentiation, and the gene expression of these markers was found to be higher in BMAds than in hBMSCs.” However, these results are not reported. They should be added.

Author’s reply: As requested by Reviewer 3, we have deleted the part on basic research.

8- Reporting only the gene expression of three markers is not enough to demonstrate that adding hSOST has no impact on the adipogenic differentiation. Other markers should be added, including protein quantification.

Author’s reply: As requested by Reviewer 3, we have deleted the part on basic research.

9- Subsections should be deleted from the discussion.

Author’s reply: we have edited the discussion.

10- The conclusions should be improved reporting not only what the authors did not find but also the results found.

Author’s reply: we have edited the conclusion.

11- “In the FRAME study, transiliac bone biopsies were performed at 2 or 12 months of treatment with romosozumab or placebo to assess BMAT parameters (43). We are look-ing forward to the results, which are expected to be published shortly.” This part should be deleted from the conclusion as it is not clear.

Author’s reply: we have edited the conclusion.

12- Line numbers should be added.

Author’s reply: we have edited the manuscript

13- The authors are encouraged to follow the guidelines of the journal. References should be checked and reported between [] and not (). Section and subsections should be numbered. Table should be revised following the layout of the journal.

Author’s reply: we have edited the manuscript according to the guidelines of the journal.

Round 2

Reviewer 1 Report

The authors answered the clinical questions I submitted. As the in vitro work has now been removed, there is nothing further to comment on.

Reviewer 3 Report

No additional comments.